



# 1 A parameterization of sulfuric acid-dimethylamine nucleation
# 2 and its application in three-dimensional modeling

Yuyang Li[1, #], Jiewen Shen[1, 2, #], Bin Zhao[1, 2, *], Runlong Cai[3], Shuxiao Wang[1, 2], Yang Gao[4], Manish Shrivastava[5], Da Gao[1, 2],
Jun Zheng[6], Markku Kulmala[2, 7, 8], Jingkun Jiang[1, *]
[1]State Key Joint Laboratory of Environment Simulation and Pollution Control, School of Environment, Tsinghua University,
100084 Beijing, China
[2]State Environmental Protection Key Laboratory of Sources and Control of Air Pollution Complex, Beijing, 100084, China
[3]Institute for Atmospheric and Earth System Research/Physics, Faculty of Science, University of Helsinki, 00014 Helsinki,
Finland
[4]Key Laboratory of Marine Environment and Ecology, Ministry of Education, Ocean University of China, Qingdao 266100,
China
[5]Brian Gaudet, Pacific Northwest National Laboratory, Richland, Washington, USA
[6]School of Environmental Science and Engineering, Nanjing University of Information Science & Technology, Nanjing
210044, China
[7]Aerosol and Haze Laboratory, Beijing Advanced Innovation Center for Soft Matter Science and Engineering, Beijing
University of Chemical Technology, 100029 Beijing, China
[8]Joint International Research Laboratory of Atmospheric and Earth System Sciences, School of Atmospheric Sciences,
Nanjing University, Nanjing, China
[#] These authors contributed equally
[*] *Correspondence to*: Bin Zhao (bzhao@mail.tsinghua.edu.cn) and Jingkun Jiang (jiangjk@tsinghua.edu.cn)





**Abstract**. Sulfuric acid (SA) is a governing gaseous precursor for atmospheric new particle formation (NPF) in diverse environments, which is a major source of global ultrafine particles. In polluted urban atmosphere with high condensation sink (CS), the formation of stable SA-amine clusters, such as SA-DMA clusters, usually initializes intense NPF events. Coagulation scavenging and cluster evaporation are dominant sink processes of SA-amine clusters in urban atmosphere, yet they are not quantitatively included in the present parameterizations of SA-amine nucleation. We herein report a parameterization of SA-DMA nucleation based on cluster dynamic simulations and quantum chemistry calculations, with certain simplifications to largely reduce the computational costs. Compared with previous SA-DMA nucleation parameterizations, this new parameterization would be able to reproduce the dependences of particle formation rates on temperature and CS. We then incorporated it in a three-dimensional chemical transport model to simulate the evolution of particle number size distributions. Simulation results show good consistency with the observations in the occurrence of NPF events and particle number size distributions in wintertime Beijing, showing a significant improvement compared to that using parameterization without coagulation scavenging. Quantitative analysis shows that SA-DMA nucleation contributes majorly to nucleation rates and aerosol population during the 3-D simulations in Beijing (>99% and >60%, respectively). These results broaden the understanding of NPF in urban atmospheres and stress the necessity of including the effects of coagulation scavenging and cluster stability in simulating SA-DMA nucleation in three-dimensional simulations. This would improve the performance in particle source apportionment and quantification of aerosol effects on air quality, human health, and climate.



## 1 Introduction

New particle formation (NPF) is the major source of atmospheric particles in terms of their number concentration, which regulates the Earth's radiative balance and affects the climate (Kulmala et al., 2004; Gordon et al., 2017; Merikanto et al., 2009). The transformation from gaseous precursors to stable clusters and particles via nucleation is the initial step of NPF, and new particle formation rate ($J$) is an essential parameter to characterize NPF intensity (Kulmala, 2003). Although nucleation processes would be suppressed by coagulation scavenging in urban atmospheres with high condensation sink (CS) (Cai and Jiang, 2017; Cai et al., 2017b), intense NPF events have been frequently observed (Wu et al., 2007; Xiao et al., 2015; Deng et al., 2020). Recently, increasing evidence has been provided that those intense events are driven by the formation of stable SA-amine clusters (Cai et al., 2022; Jen et al., 2014b) with a speed close to the collision limit for SA molecules, thus deriving high nucleation rates in urban atmospheres (Cai et al., 2021d; Yao et al., 2018; Chen et al., 2012). Thus, integrating SA-amine nucleation into three-dimensional (3-D) models would be essential in extending the understanding of NPF in polluted urban areas and quantifying its underlying impacts on the environment and climate. This requires a quantitative representation of particle formation rates through SA-amine nucleation for 3-D models.

Semi-empirical power-law functions are widely used in SA-relevant nucleation rate studies to fit the experimental data, which has been shown to reproduce the measured $J$ in certain ambient observations or experimental conditions (Riccobono et al., 2014; Dunne et al., 2016; Bergman et al., 2015; Hanson et al., 2017; Semeniuk and Dastoor, 2018; Kurten et al., 2014; Kurten et al., 2018). For SA-amine nucleation, Bergman et al. (2015) and Dunne et al. (2016) have presented semi-empirical parameterizations of good consistencies with chamber and flow-tube experimental results (Almeida et al., 2013; Jen et al., 2014b; Glasoe et al., 2015). In real urban atmosphere, recent advances have shown that coagulation scavenging would largely suppress concentrations of molecular clusters, and thus the nucleation rates (Cai and Jiang, 2017; Cai et al., 2021c; Cai et al., 2021d; Marten et al., 2022). It has also been addressed that the formation of the smallest SA-amine clusters, which is largely dependent on cluster stability, is the limiting step for SA-amine nucleation rates (Cai et al., 2022). However, the effects of coagulation scavenging and cluster stability would vary with the environmental factors, e.g., CS and temperature, while these effects have not been well represented in semi-empirical power-law functions derived from certain experimental systems or ambient environments. Cluster kinetic simulations coupled with quantum chemistry calculations (Mcgrath et al., 2012), which take into account the effects of both coagulation scavenging and cluster stability, have been widely applied in zero-dimensional or one-dimensional simulations of SA-NH$_3$ or SA-amine nucleation (Yang et al., 2021; Lu et al., 2020; Yao et al., 2018; Yu, 2006; Yu and Turco, 2001). Specifically, both cluster kinetic simulations and observations reveal that dimethylamine (DMA) is plausibly most efficient in stabilizing SA clusters and is regarded as the key amine species deriving high particle formation rates in urban atmosphere (Jen et al., 2014b; Cai et al., 2022; Yao et al., 2018; Chen et al., 2012). However, no method with good representations of coagulation scavenging and cluster stabilities has been reported to explicitly simulate the SA-DMA nucleation rates in 3-D chemical transport models.

A challenge in setting up a parameterization based on cluster kinetic simulations for 3-D chemical transport models is to reduce computational costs and yield explicit expressions. A plausible method to reduce computational costs is to omit the unstable clusters with high evaporation rates from the nucleation pathway. Accordingly, different nucleation schemes were presented to represent the dominant source or sink processes of SA-DMA clusters in specific chamber experiments or ambient environments (Lu et al., 2020; Cai et al., 2021d). For polluted urban atmospheres, a kinetic model with a key pathway of particle formation in SA-DMA nucleation was constructed, yielding good predictions of measured SA cluster concentrations and 1.4 nm particle formation rates ($J_{1.4}$) in urban Beijing (Cai et al., 2021d). Application of pseudo-steady-state assumptions is also an alternative method for reducing computational costs and yielding explicit expressions. The NPF occurrence indicator ($I$) based on the kinetic model with pseudo-steady-state assumptions has shown good consistency in qualitatively estimating the NPF events in urban Beijing and Shanghai (Cai et al., 2021c). These results indicate the potential of deriving an explicit





parameterization of particle formation rates by applying pseudo-steady-state assumptions to the kinetic model, although further
quantitative analysis is still required to validate this parameterization.
In this study, we set up an SA-DMA nucleation parameterization, which is designed for application in 3-D chemical transport
models. The parameterization is based on the pseudo-steady-state particle formation rate in the kinetic model, with a full
representative of the effects of coagulation scavenging and cluster stability (Cai et al., 2021d). Generally, only four variables
(temperature $T$, CS, gaseous DMA concentrations $[B]$, and concentrations of SA molecules or clusters containing one SA
molecule $[SA_{tot}]$) are used in the parameterization, with computational costs largely reduced. We then implement the
parameterization in a 3-D chemical transport model and combine it with an integrated source-sink representation of DMA to
simulate the evolution of the particle number size distributions (PNSDs) in wintertime Beijing. The precursor concentrations,
PNSDs, NPF occurrence and $J_{1.4}$ show relatively good consistencies between simulations and observations. The simulations
show that the SA-DMA nucleation contributes >99% of the $J_{1.4}$ and >60% of the total particle number concentration in
wintertime Beijing, respectively. With this parameterization, 3-D chemical transport models could significantly improve the
simulation of NPF, especially in urban environments, and thus the effects of NPF on particulate matter pollution or climate.

## 2 Methods

### 2.1 Derivation of Parameterized Formation Rate in SA-DMA Nucleation

Limited by computational quantum chemistry calculation results, SA-DMA nucleation is commonly simulated in the range of
clusters containing not more than 4 SA or 4 DMA molecules (Olenius et al., 2013; Ortega et al., 2012). As unstable clusters
would evaporate with higher rates, the formation of larger clusters potentially follows the pathways of the most stable clusters.
In addition, as the SA-DMA clusters are increasingly stable along the main pathway of cluster formation, the clusters not
smaller than $A_4B_4$ (hereafter $A_mB_n$ refers to clusters containing m SA and n DMA molecules) is assumed to not evaporate back
in these simulations. Although there are uncertainties in the pathways presented based on different quantum chemistry methods,
it is well accepted that the $A_mB_m$ (m=1 to 4) and $A_2B_1$ clusters are relatively stable in the SA-DMA nucleation scheme (Olenius
et al., 2017; Olenius et al., 2013; Ortega et al., 2012; Myllys et al., 2019).
Accordingly, the parameterization in this study is derived from the nucleation pathway including $A$, $B$ and other 5 SA-DMA
clusters ($A_mB_m$ (m=1 to 4) and $A_2B_1$), consistent with a previous study (Cai et al., 2021d). The clusters except $A_4B_4$ are assumed
to be in pseudo-steady-states, i.e. the sink due to evaporation, coagulation scavenging, and cluster collision is equal to the
source due to the collisions of molecules or smaller clusters. As the $A_4B_4$ clusters are estimated to be with an electrical mobility
diameter of approximately 1.4 nm, the pseudo-steady-state formation rate of $A_4B_4$ would be applied in the parameterization of
$J_{1.4}$ in this study.

### 2.1.1 Derivation of Collision Coefficients, Coagulation Sink, and Evaporation Rates

In the nucleation pathway discussed above, $A$, $B$, and 5 SA-DMA clusters are included. The collision coefficients between
them ($\beta_{i-j}$) and the evaporation rate of $A_1B_1$ clusters ($\gamma$) would vary with $T$ during the simulation. The coagulation sinks (CoagS$_i$)
due to the coagulation scavenging of background aerosols are dependent on CS. The work discussed in this section is focused
on simplification of the derivation of these parameters to be updated in each simulation time interval to reduce the
computational costs.
As the involved clusters and molecules are in the free molecular regime (Knudsen number > 10), $\beta_{i-j}$ in SA-DMA nucleation
processes can be calculated based on kinetic gas theory (Seinfeld and Pandis, 1998; Olenius et al., 2013; Ortega et al., 2012):



$$\beta_{i\text{-}j} = \left(\frac{3}{4\pi}\right)^{1/6}\left(\frac{1}{m_i}+\frac{1}{m_j}\right)^{1/2}(V_i^{1/3}+V_j^{1/3})^2(6k_{\mathrm{b}}T)^{1/2}E_{ij}, \tag{1}$$

where $m$ (kg) and $V$ (m³) represent the molecular mass and molecular volume, respectively. The density of precursor molecules $A$ and $B$ was assumed to be 1830 and 680 kg m⁻³, respectively. $T$ (K) represents the temperature. $k_{\mathrm{b}}$ (J K⁻¹) is the Boltzmann constant. Subscripts $i$ and $j$ refer to the index of the clusters or molecules (1 to 7 refer to $A$, $B$, $A_1B_1$, $A_2B_1$, $A_2B_2$, $A_3B_3$, and $A_4B_4$, respectively, which are involved in the kinetic model). $E_{ij}$ is a dimensionless enhancement factor of the collision rates from Van de Waals forces between $i$ and $j$. In this study, $E_{ij}$ is assumed to be 2.3 (Chan and Mozurkewich, 2001; Sceats, 1989), within the range of 2.3 to 2.7 predicted by Brownian coagulation models, and consistent with the value used in other cluster dynamics studies (Kurten et al., 2014; Lehtipalo et al., 2016; Stolzenburg et al., 2020).

Noting that $m_i$ and $V_i$ are almost independent of the atmospheric conditions and $E_{ij}$ is assumed to be constant, we can normalize different values of $\beta_{i\text{-}j}$ into $\beta$, and the normalizing factor is shown in a look-up table (Table S1 in the supporting information (*SI*)) as $G(i,j)$:

$$\beta_{i\text{-}j} = \beta G(i, j), \tag{2}$$

where $\beta$ represents the collision coefficients between two $A_1B_1$ clusters ($\beta_{3\text{-}3}$), and could be calculated as:

$$\beta = \beta_0\left(\frac{T}{T_0}\right)^{0.5}, \tag{3}$$

where $\beta_0$ is the value of $\beta$ at the standard temperature $T_0$=298.15 K, constant as $1.126\times10^{-15}$ m³ s⁻¹.

Similarly, CoagS$_i$ could also be normalized to CS using fixed ratios. The size dependent coagulation sink (CoagS) is calculated with a power-law exponent of -1.7, within the typical range of atmospheric aerosols (Lehtinen et al., 2007):

$$CoagS_i = CS\left(\frac{V_i}{V_1}\right)^{\frac{1.7}{3}}=H(i)CS, \tag{4}$$

where the dimensionless factors $H(i)$ are also recorded in Table S1 in the *SI*.

The evaporation rates of $A_1B_1$ could be derived based on collision-evaporation equilibrium (Ortega et al., 2012), closely relevant to the free energy barrier to form $A_1B_1$ clusters (Olenius et al., 2013; Ortega et al., 2012):

$$\gamma = \beta_{1\text{-}2}c_{\mathrm{ref}}\exp\left(\frac{\Delta G}{k_B T}\right), \tag{5}$$

where $c_{\mathrm{ref}}$ is the number concentrations under standard conditions ($2.46\times10^{25}$ m⁻³). $\Delta G$ is the formation free energies of $A_1B_1$.

Thus if we take $T_0 = 298.15$ as a reference, $\gamma$ could also be calculated as:

$$\gamma = \gamma_0\left(\frac{T}{T_0}\right)^{0.5}\exp\left(\frac{\Delta H}{k_{\mathrm{B}}}\left(\frac{1}{T}-\frac{1}{T_0}\right)\right), \tag{6}$$

$$\gamma_0 = \gamma_0'\exp\left(\frac{\Delta G-\Delta G_0}{k_{\mathrm{B}}T_0}\right), \tag{7}$$

where $\gamma_0'$, with the value of 3.33 s⁻¹, is the evaporation rates of $A_1B_1$ at $T_0$ with $\Delta G=\Delta G_0$=-13.54 kcal mol⁻¹. $\Delta H$ is the formation enthalpies of $A_1B_1$. In previous studies, several sets of $\Delta H$ and $\Delta G$ at specific temperatures were reported based on different quantum chemistry models. Here we use $\Delta H$ = -24.82 kcal mol⁻¹ and $\Delta G$ = -13.54 kcal mol⁻¹ according to the results in Myllys et al. (2019). If the values of $\Delta G$ need to be updated in future application of this parameterization, the values of $\gamma_0$ should be updated as well based on Eq. 7. The sensitivity analysis of different values of $\Delta H$ and $\Delta G$ are discussed in the Results section.

Generally, with $G(i,j)$ and $H(i)$ fixed into the parameterization formula, $\beta_{i\text{-}j}$ and CoagS$_i$ could be normalized to $\beta$ and CS. Additionally, the values of $\gamma$ and $\beta$ could be real-time updated at any simulation timestep based on Eqs. 3 and 6.

**2.1.2 Formula of the SA-DMA Nucleation Parameterization**

Applying the pseudo-steady-state assumptions to the key pathway discussed above (Eqs. S1 to S9) and achieving real-time $\gamma$ (s⁻¹) and $\beta$ (m³ s⁻¹) (Eqs. 3 and 6), we could derive an explicit formula of the parameterized $J_{1.4}$ in this study (Eq. 8).



$$J_{1.4} = \frac{\beta\theta'[A_1B_1]^4}{2([A_1B_1]+0.39\frac{CS}{\beta})}\left(\frac{0.23\theta'}{[A_1B_1]+0.39\frac{CS}{\beta}}+\frac{1.00}{[A_1B_1]+0.31\frac{CS}{\beta}}\right),$$ (8)
The above intermediate parameters are calculated as below:
$$[A_1B_1] = \frac{0.96[B][SA_{tot}]}{0.96[B]+\frac{\gamma}{\beta}+0.86[SA_{tot}]+0.63\frac{CS}{\beta}},$$ (9)
$$\theta = 1+\frac{2[B]}{1.16[B]+0.46\frac{CS}{\beta}}\frac{[SA_{tot}]-[A_1B_1]}{[A_1B_1]},$$ (10)
$$\theta' = \frac{\theta(2.22[A_1B_1]+0.86\frac{CS}{\beta})}{\sqrt{(1.11[A_1B_1]+0.43\frac{CS}{\beta})^2+1.12\theta[A_1B_1]^2+1.11[A_1B_1]+0.43\frac{CS}{\beta}}},$$ (11)
In Eqs. 8 to 11, the four input variables ($T$ (K), CS ($s^{-1}$), $[B]$ ($m^{-3}$), $[SA_{tot}]$ ($m^{-3}$)) are shown in bold. Generally, only these four
variable parameters are needed for the 3-D chemical transport models. Additionally, compared with directly coupling cluster
dynamic simulations into 3-D chemical transport models, the parameterization of pseudo-steady-state $J_{1.4}$ requires much less
computational time.

### 2.2 Incorporating the Parameterization into Updated WRF-Chem/R2D-VBS Model

The updated parameterization of SA-DMA nucleation was incorporated in the WRF-Chem (Weather Research and Forecasting
model with Chemistry). Before adding the SA-DMA nucleation, we already incorporated seven other NPF mechanisms in the
model (Zhao et al., 2020):  four inorganic pathways, including binary neutral/ion-induced SA-H$_2$O nucleation and ternary
neutral/ion-induced NH$_3$-SA-H$_2$O nucleation; and three organic pathways, including pure-organic neutral/ion-induced organic
nucleation and ternary nucleation involving organics and SA. The organic containing nucleation pathways are driven by ultra-
and extremely low volatility organic compounds (ULVOC and ELVOC) with O:C > 0.4, converted from monoterpene
autoxidation. The chemical transformation and volatility distribution of monoterpene is represented in the model by R2D-VBS
(Radical Two-Dimensional Volatility Basis Set framework) with constrained parameters against experiments. More details of
the R2D-VBS are given in our previous study (Zhao et al., 2020). The newly formed nano-sized particles and their initial size
evolution are accounted in the MOSAIC module by 20 size bins covering 1 nm to 10 μm. It is worth mentioning that the newly
formed particles from SA-DMA nucleation are lumped into a lower aerosol size bin in the model than that of other seven
pathways.  This should be attributed to that our SA-DMA nucleation parameterization are formulated at a 1.4 nm-sized particle
formation rate while the remaining ones are fitted based on measured particle formation rates from CLOUD Chamber at a
mobility diameter of 1.7 nm. Given that condensation of gaseous SA and DMA on pre-existing aerosols and nucleation occur
simultaneously in real atmosphere, in the model, we then use a time-integrated-averaged concentration of precursors over each
time step to drive SA-DMA nucleation. The condensation sink for SA and DMA is calculated according to simulated real-
time PNSDs. In addition, the consumption of both SA and DMA concentration during nucleation is also accounted in the
model, in order to represent a comprehensive sources-sink simulation scheme of two precursors in combination with other
settings.

### 2.2.1 Sources and Sinks of Dimethylamine in the Updated WRF-Chem/R2D-VBS Model

A regional or global bottom-up emission inventory of DMA is currently lacking, mostly due to scarce direct measurements
(Yang et al., 2022; Zhu et al., 2022). In previous 3D model studies, amine/NH$_3$ emission ratios have often been used to estimate
amine emissions due to the close correlation between NH$_3$ and DMA emissions. However, a fixed amine/NH$_3$ ratio is likely
to overestimate the concentrations of amines in rural areas while underestimating those in urban areas, where high
concentrations of amines have been reported (Yao et al., 2018; Bergman et al., 2015). Here, a set of source-dependent





DMA/NH$_3$ emission ratio was used to develop the emission inventory of DMA based on (Mao et al., 2018). The ratios for
different emission sectors were determined by a source apportionment analysis, based on a simultaneous observation of NH$_3$,
C1-C3 amines, NO$_x$, and SO$_2$ and also meteorological factors at a suburban site in Nanjing (Zheng et al., 2015a). We applied
the source-dependent emission ratios (0.0070, 0.0018, 0.0015, 0.0100, and 0.0009 for chemical–industrial, other industrial,
agricultural, residential, and transportation source types, respectively) to NH$_3$ emissions in the ABaCAS-EI 2017 (Emission
Inventory of Air Benefit and Cost and Attainment Assessment System) for China mainland and the IIASA 2015 emission
inventory for other areas to build continental DMA emission inventory (Zheng et al., 2019; Gao et al., 2020). In addition,
DMA emission for maritime area was developed employing a DMA/NH$_3$ ratio derived from recent campaigns in offshore
areas of China (see details in SI) (Chen et al., 2021).
DMA can be removed from the atmosphere through three main pathways: gas-phase chemical reaction, aerosol uptake, and
wet deposition, which are all explicitly considered in our model. For the gas-phase chemical reactions, only oxidation of DMA
by •OH is included. Reactions with other oxidants (O$_3$ and NO$_3$) are much slower and therefore have negligible effects on
DMA concentrations (Ge et al., 2011). The mechanism of DMA concentration depletion by aerosol uptake is still poorly
understood, and the key parameter, uptake coefficient $\gamma_u$, varies in a wide range depending on many factors such as aerosol
composition and relative humidity. In this study, we assumed $\gamma_u = 0.001$, approximately a median value among those reported
by recent laboratory measurements (Qiu et al., 2011; Wang et al., 2010). Regarding DMA depletion by wet deposition, the
treatment is similar to that of NH$_3$ based on Henry's Law. The key parameters for above sink processes are summarized in
Table S2 in the *SI*.

### 2.2.2 Configuration of the Updated WRF-Chem/R2D-VBS Model.

The WRF-Chem model configured with the SA-DMA nucleation is applied to a domain covering eastern Asia with a horizontal
resolution of 27 km, where Beijing is located close to the center. The simulations are performed for two winter months
separately (December 2018 and January 2019) with 5 days spin-up run for each month. The ABaCAS-EI 2017 and IIASA
2015 emission inventory were used for China mainland and other areas, respectively. The biogenic emission is calculated by
the Model of Emissions of Gases and Aerosols from Nature (MEGAN) v2.04 (Guenther et al., 2006). Except for the
monoterpene-related gas and aerosol chemistry that is traced by R2D-VBS, the remaining gas- and aerosol chemical processes
are simulated by the SAPRC99 gas chemistry scheme coupled with the MOSAIC (Model for Simulating Aerosol Interaction
and Chemistry) aerosol module and a one-dimensional VBS set for SOA modeling (Zaveri et al., 2014; Shrivastava et al.,
2019; Shrivastava et al., 2011).
Four scenario simulations with different configurations of the NPF mechanisms were conducted in this study to examine how
the SA-DMA nucleation affects the simulations of aerosol size distribution: 1) 8 NPF mechanisms with the SA-DMA
nucleation rate at 1.4 nm (abbr. DMA1.4_Mech8); 2) 8 NPF mechanisms with the SA-DMA nucleation rate at 1.7 nm
converted using modified Kerminen-Kulmala equation (Lehtinen et al., 2007) (DMA1.7_Mech8); 3) 7 NPF mechanisms
without the SA-DMA nucleation (NoDMA_Mech7); and 4) No NPF mechanism (NoDMA_Mech0). Among them, scenario 1
is our "best-case" with a full consideration of available nucleation mechanisms; scenario 2 is set to probe the feasibility to use
modified Kerminen-Kulmala equation to simulate the initial particle growth; scenario 3 is the "base-case" representing the
performance of the original model; and scenario 4 represents the evolution of aerosol population only contributed by primary
emission. Scenarios 3 and 4 were set as controlling groups to assess the role of SA-DMA nucleation and other mechanisms.

### 2.3 Ambient Measurements

Ambient observations were conducted at an urban site in Beijing from January 2018 to April 2018 and from October 2018 to
March 2019. The site is located on the West Campus of Beijing University of Chemical Technology. Details of the observation
site can be found in previous studies (Liu et al., 2020; Deng et al., 2020). The concentrations of SA and involving clusters are





measured using a chemical ionization high resolution time of flight mass spectrometer (CI-HTOF-MS) and a chemical
ionization time of flight mass spectrometer with a long mass analyzer (CI-LTOF-MS) (Bertram et al., 2011; Jokinen et al.,
2012). Other details in the sampling configurations have been reported in our previous study (Deng et al., 2020). Amine
concentrations are measured using a modified time of flight mass spectrometer (TOF-MS) (Zheng et al., 2015b; Cai et al.,
2021b). A weather station was deployed to measure the meteorological data, including ambient temperature, relative humidity
and pressure. The PNSDs of particles from 1 nm to 10 μm were measured using a particle size distribution (PSD) and a diethyl
glycol-scanning mobility particle sizer (DEG-SMPS) (Jiang et al., 2011; Liu et al., 2016; Cai et al., 2017a). CS is calculated
from the measured PNSDs and $J_{1.4}$ is calculated using an improved aerosol population balance formula (Cai and Jiang, 2017).
The details of instrument calibrations and data validations can be found in our previous study (Cai et al., 2021b).
**3 Results and discussion**
**3.1 Validation of Parameterization**
The reasonability of pseudo-steady-state assumptions in the SA-DMA nucleation pathway was tested through comparisons
between the characteristic equilibrium time (τ) of kinetic simulation (see details in the *SI*) and the data collection time interval.
The characteristic equilibrium time of involving clusters and simulated $J_{1.4}$ were shown in Fig. S1 in the *SI*. Generally, in either
clean and cold circumstances or polluted and warm circumstances, the kinetically simulated $J_{1.4}$ could be well reproduced by
parameterized pseudo-steady-state $J_{1.4}$. Actually, τ would vary greatly with CoagS and γ, and would be higher on cleaner and
colder days, while even in extremely clean and cold days with $CS = 0.0001$ s$^{-1}$ and $T = 255$ K, τ of $A_3B_3$ (longer than other
clusters) is only ~20 min, shorter than the data collection time interval of 30 min. Thus for circumstances where there are high
atmospheric concentrations of DMA and SA, such as most typical polluted regions, we conclude that nucleation processes are
rapid enough that kinetic $J_{1.4}$ can be represented by pseudo-steady-state $J_{1.4}$.
Figure 1 presents the comparisons between parameterized $J_{1.4}$ in this study and those simulated in the kinetic models (hereafter
referred to as KM) presented by Cai et al. (2021) and the cluster dynamic simulations containing all $A_mB_n$ (m, n≤4) clusters
(hereafter referred to as CDS). The simulated $J_{1.4}$ in KM can be reproduced by parameterized $J_{1.4}$ within a ±50% range for
most of the cases in urban Beijing, with no systematic deviations found between them.
Figure 1b shows that for most of the circumstances, deviations between the parameterized $J_{1.4}$ and $J_{1.4}$ simulated in CDS are
within a range of 1 order of magnitude. However, for circumstances with high temperatures, the parameterized $J_{1.4}$ would be
higher than those simulated in CDS, which might be due to that the $A_kB_k$ (k=2,3 and 4) clusters are assumed to be non-
evaporative in KM while they would evaporate back in CDS under high temperatures. The reasonability of cluster stability
assumptions under high temperatures relies mainly on the accuracy of quantum chemistry calculations, which requires more
experimental evidence and discussions. Additionally, due to the negative dependence of simulated $J_{1.4}$ on $T$, the simulated $J_{1.4}$
in this parameterization would be mostly lower than 10 cm$^{-3}$s$^{-1}$ under temperatures higher than 15 ℃, lower than the median
and mean value of particle formation rates measured during long-term observations in Beijing (Deng et al., 2021). Although
they are relatively higher than those simulated in CDS, the simulation results of NPF occurrence would not show large
deviations.
The computational costs of these three simulations have also been tested on the same personal computer with a Matlab program.
To achieve the steady-state $J_{1.4}$ in a specific atmospheric condition, the CDS and KM needs ~10 s and ~0.05 s CPU time,
respectively, while the calculation of parameterized pseudo-steady-state $J_{1.4}$ merely costs ~2×10$^{-7}$ s CPU time. The CPU time
was reduced by a factor of 5×10$^7$ and 4×10$^4$ compared to CDS and KM, respectively. Thus introducing this parameterization
into 3-D chemical transport models could largely reduce the computational costs.





**3.2 The Dependence of Parameterized $J_{1.4}$ on Input Parameters**
The correlation between parameterized SA-DMA nucleation $J_{1.4}$ and the input parameters are shown in Fig. 2. The parameters
involved are $T$, CS, [DMA], and [$SA_{tot}$]. The mean values of measured data during the observation period (281K, 0.02 s$^{-1}$, 3
ppt, and $3.5 \times 10^6$ cm$^{-3}$, respectively) are applied as typical conditions in the base case. Different from the semi-empirical power-
law functions only based on precursor concentrations presented by Dunne et al. (2016), the dependences of particle formation
rates on $T$ and CS are represented in our parameterizations. With $T$ increasing from -10 to 20 ℃, $\gamma$ would increase by ~2 orders
of magnitude, as shown in Fig. 2a, and thus $J_{1.4}$ would decrease by over 2 orders of magnitude. The decreasing trend of
observed NPF rate ($J_{1.5}$ in this case) as a function of increasing $T$ in urban Beijing has also been reported (Deng et al., 2020),
consistent with our parameterizations.
Fig. 2b shows that $J_{1.4}$ would decrease by 2-4 orders of magnitude with CS increasing by a factor of 10, and the logarithm
dependence is higher in circumstances with higher CS, such as urban Beijing, where CoagS dominates the sinks. This is
consistent with the negative CS dependence of measured particle formation rates and NPF occurrence demonstrated in previous
observations in Beijing (Deng et al., 2021; Cai et al., 2021b; Cai et al., 2021a).
The parameterized $J_{1.4}$ shows an increasing trend with increasing concentrations of SA and DMA. Parameterized $J_{1.4}$ is
approximately proportional to [SA]$^4$, while the dependence of $J_{1.4}$ on [DMA] is decreasing with increasing [DMA]. This is
due to the near-saturation formation of $A_1B_1$ clusters, which is also found in kinetic model simulation results (Cai et al., 2021d).
Generally, the parameterization could reproduce the fact that SA-DMA nucleation is driven by SA-DMA cluster formation,
dominantly suppressed by cluster evaporation and coagulation sinks.
**3.3 Comparison of 3D Model Simulations with Observations**
As DMA and SA concentrations are key input variables for the SA-DMA nucleation parameterization, we first compare
simulated DMA and SA concentrations from the DMA1.4_Mech8 scenario with observations (Fig. 3). Generally, there are
good consistencies of both mean concentrations and temporal variations, although there are still deviations at certain times.
The mean simulated concentrations of DMA and SA are 1.9 ppt and $1.4 \times 10^6$ cm$^{-3}$, respectively, close to observed
concentrations of 2.0 ppt and $1.6 \times 10^6$ cm$^{-3}$. This proves the validity of the comprehensive representation of source-sink
behaviors of DMA in the model.
The time series of PNSDs for different simulation scenarios are presented in Fig. 4. When SA-DMA nucleation is considered,
the typical PNSDs shape in observed NPF days (12/07, 12/08, 12/09, 01/20, and 01/21), characterized as the burst of
nanometer-sized particles and subsequent growth, are well captured by our "best-case" scenario DMA1.4_Mech8 and also
DMA1.7_Mech8. By contrast, the scenarios without DMA-SA nucleation, NoDMA_Mech7 and NoDMA_Mech0, cannot
reproduce the observed NPF events with a "vacancy band" for 1~10 nm size range over the entire simulation period. Actually,
although there are slightly higher sub-3 nm particle concentrations in NoDMA_Mech7 than those in NoDMA_Mech0, which
are generated from the 7 nucleation pathways other than DMA-SA nucleation, the newly formed particle concentrations are
too low to survive in the subsequent growth and be separated from background aerosols in the PNSDs. These results
demonstrate that SA-DMA nucleation should be the dominant mechanism during NPF events in Beijing compared with other
7 mechanisms.
Our results also reproduce the dependence of NPF occurrence on CS in Beijing. As shown in Fig. S2 in the *SI*, NPF generally
occurs at low CS while high CS results in too low nucleation rates to initiate NPF. Note that the simulated sub-3 nm particle
concentrations also increase slightly on some non-NPF days in DMA1.4_Mech8 and DMA1.7_Mech8 scenarios, however,
the concentrations are ~1 order of magnitude lower than those on NPF days and the newly formed particles also fail to survive
in the subsequent growth. The improvements of using the nucleation parameterization in this study is further stressed in the





comparison between DMA1.4_Mech8 scenario and the scenario (CLOUD) using the parameterization from Dunne et al.
(2016). Figure S3 has shown that almost no rapid nucleation processes and NPF events are found in the simulation of CLOUD
scenarios. In addition to the underestimation of nucleation rates, the simulated high nucleation rates usually occur on observed
non-NPF days (Fig.S6), which should be attributed to the ignorance of CS dependence in the power-law function
parameterizations.
Figure 5 further compares the simulated and observed PNSDs averaged over the simulation period. The "best-case" scenario
DMA1.4_Mech8 brings the averaged PNSD in 1~200 nm size range much closer to the observation than those of "base-case"
NoDMA_Mech7, and the latter only shows a minor change compared to scenario NoDMA_Mech0 without any nucleation.
One may notice that the averaged PNSD in 2~10 nm size range for scenario DMA1.4_Mech8 is still lower than that of
observation by ~1 order of magnitude, despite the good agreement in number concentrations of particles of ~1.4 nm. This
could be attributed to two possible reasons: the model underestimates the actual nucleation rates; or newly formed particles of
~1.4 nm grow too fast to larger size bins in the model (> 10 nm). The first one can be excluded by a generally good agreement
between simulated nucleation rates and ones derived from observation, even with a slightly higher mean value for the former
(shown in next section, Fig. 6). Hence, the gap in 2~10 nm size range might be attributed to the particle growth simulations in
the model which deserves further improvement. Moreover, in spite of similar performance in improving PNSDs simulations
compared to the "best-case" DMA1.4_Mech8, the scenario of DMA1.7_Mech8 presents a shifted PNSD pattern to larger size
range. For these two scenarios including SA-DMA nucleation, scenario DMA1.4_Mech8 is more reasonable since a systematic
underestimation exists over the entire 1~10 nm range in scenario DMA1.7_Mech8. Still, the conversion from 1.4 nm rate to
those for larger particles through modified Kerminen-Kulmala equation is an alternative way to depict SA-DMA nucleation
for other models with different aerosol size settings.Overall, despite aforementioned deficiencies, our updated WRF-
Chem/R2D-VBS model configured with the SA-DMA nucleation parameterization shows substantial improvement in
representation of NPF events and the PNSD.

### 3.4 Contribution from Various Pathways to Nucleation Rates and Particle Number Concentrations

Quantitative analysis over various nucleation pathways is performed here to improve the understanding of NPF in Beijing. As
presented in Fig. 6, the variation of nucleation rates, which are derived from observed PNSD data, is well represented by the
best-case scenario DMA1.4_Mech8. Compared to the vast majority contribution from SA-DMA nucleation, the nucleation
rates from other nucleation mechanisms are lower by a factor of ~100. In addition, SA-DMA nucleation contributes over 60%
to aerosol population, reinforcing its dominant role in modulating aerosol population in urban atmosphere.

### 3.5 Sensitivity Analysis

Having shown the significant improvement of model performance in simulating NPF by coupling the SA-DMA nucleation
parameterization, we acknowledge that the simulation of SA-DMA nucleation in 3D model still has uncertainties in terms of
both source-sink representation of DMA and nucleation parameterization. Here, several key factors which may alter model
performance were selected to perform sensitivity analysis.
First, the uncertainties brought by $\Delta G$ achieved from different quantum chemistry results are tested for both parameterized $J_{1.4}$
and the 3-D chemical transport model simulations. In previous studies, a number of $\Delta G$ values have been reported: -11.02
kcal mol$^{-1}$ (Ge et al., 2020), -15.40 kcal mol$^{-1}$ (Ortega et al., 2012), -13.54 kcal mol$^{-1}$ (Myllys et al., 2019). The $\Delta G$ of -14.00
kcal mol$^{-1}$ was applied in (Cai et al., 2021d) to achieve good consistencies between simulated and measured $J_{1.4}$ is also applied
in the sensitivity analysis. Figure S7 shows the variation of parameterized $J_{1.4}$ applying different $\Delta G$ values at 281 K, the
median temperature of the observation period. For DMA with median values of ~3 ppt, different $J_{1.4}$ could vary by ~5 orders
of magnitude with $\Delta G$ between -11.02 kcal mol$^{-1}$ and -15.40 kcal mol$^{-1}$, while $J_{1.4}$ with $\Delta G$ of -13.54 kcal mol$^{-1}$ is also lower





than that of -15.40 kcal mol$^{-1}$ by a factor of ~10. However, if the DMA concentrations are up to ~30 ppt, the differences of $J_{1.4}$
when $\Delta G$ varies between -13.54 kcal mol$^{-1}$ and -15.40 kcal mol$^{-1}$ would become much smaller, due to the saturated formation
of $A_1B_1$ clusters. For the temperature of 298.15 K, the sensitivities of parameterized $J_{1.4}$ are relatively larger, because the
formation of $A_1B_1$ clusters is far from saturation. Generally, the parameterized $J_{1.4}$ could be very sensitive to different $\Delta G$
values achieved from quantum chemistry results due to the essential influence of cluster stabilities. As a result, using a lower
$\Delta G$ value of -15.40 kcal mol$^{-1}$ in the 3-D simulations with the DMA1.4_Mech8 scenario configuration could lead to much
higher nucleation rates compared to the observation (Fig. S8). Thus we call for a more systematic performance assessment of
quantum chemistry calculation methods to constrain the uncertainties of cluster thermodynamic stabilities.
Moreover, for the DMA source, we conduct two sensitivity scenarios of doubling (DMA2) and halving (DMA0.5) the inputted
DMA emission to test the influence of limited measurements in constraining the DMA/NH$_3$ emission ratio. As for the three
sink processes, the parameters for DMA-•OH reaction and wet deposition reported in the literature have relatively small
differences while aerosol uptake coefficient of DMA covers a wide range over two orders of magnitude. We then conduct two
sensitivity scenarios using the upper ($4.4\times10^{-2}$, Upt4.4E-2) and lower ($5.9\times10^{-4}$, Upt5.9E-4) limit of aerosol uptake coefficient.
All sensitivity scenarios are on the basis of the DMA1.4_Mech8 configuration. The influence of scaled DMA emissions and
varying uptake coefficients on simulated DMA concentration, PNSDs, and nucleation rate is shown in Fig. S9-S16 in the *SI*.
As expected, the DMA concentration, especially for the nighttime spikes, is sensitive to the emission change. This causes an
overestimation for DMA2 (by a factor of ~2) and underestimation for DMA0.5 (by a factor of ~0.5), judged by monthly
averaged PNSD and nucleation rates. The sensitivity analysis for the uptake coefficient, however, shows different results. A
higher uptake coefficient of $4.4\times10^{-2}$ leads to a much lower DMA concentration (10% of the "best-case") while DMA
concentration only increase slightly when the lower limit of $5.9\times10^{-4}$ is used. Moreover, the change in uptake coefficient show
limited effect on PNSD. The reason is that the DMA concentrations during NPF periods are much less affected by the changes
in uptake coefficient than those in non-NPF periods, since NPF usually occurs at low CS conditions when the uptake of DMA
is weak.  The sensitivity analysis above show that the parameters used in our simulation are reasonable, since perturbations
within the ranges reported in the literature generally worsen the model performance. We also expect more field measurements
of DMA emission and its aerosol uptake to further constrain the key source-sink process parameters in the simulation of DMA,
although some of them show minor effect on NPF and PNSD simulations.
**4 Conclusions**
This study presents a SA-DMA nucleation parameterization for application in 3-D chemical transport models. Compared to
semi-empirical power-law fitting parameterizations, this new parameterization is based on the key pathway of SA-DMA
cluster formation and make good representations of the coagulation scavenging effect and cluster stability. Pseudo-steady-
state assumptions are applied and validated according to the short characteristic equilibrium time and through comparisons
with the cluster dynamic simulations and the kinetic model. Compared with simulating the SA-DMA nucleation with cluster
dynamic simulations or the kinetic model, applying this parameterization into 3-D chemical transport models would largely
reduce the computational costs.
We incorporate this new parameterization as well as the sources and sinks of DMA into the WRF-Chem/R2D-VBS model.
Using the updated model, we simulate the DMA concentrations and PNSDs in Beijing during December 2018 and January
2019. Comparisons are made between 3-D model simulations and ambient measurements. Good consistency is achieved in
simulating the precursor concentrations, which validates the source-sink simulation of SA and DMA. Primarily, our
quantitative analysis show that compared to other nucleation mechanisms, SA-DMA nucleation would contribute to >99% of
particle formation rates and >60% of particle number concentrations during the simulation period in urban Beijing. Although





the uncertainties exist due to the excess rapid growth in 3-D simulation, SA-DMA nucleation should be dominant sources of
aerosol population due to its dominance in new particle formation rates. Furtherly, the 3-D simulations with this
parameterization make good predictions of the CS-dependent NPF occurrence in urban Beijing and quantitatively reproduce
the particle size distributions. These demonstrate that incorporating the SA-DMA nucleation parameterization including the
effect of coagulation scavenging and cluster stabilities with 3-D chemical transport models would significantly improve the
simulation of NPF and the particle size distributions. Such improvement would be important for further simulations of cloud
condensation nuclei and the climate effects of aerosols and NPF events. The improved simulations of particle size distributions
also provide more evidence for quantitatively evaluate the environmental and health effect of ultrafine particles.
This study has emphasized that 3-D simulations with this new parameterization could reproduce the CS-dependent particle
formation rates and NPF occurrence in Beijing. As CS could vary in a relative wide range between NPF days and non-NPF
days in urban atmosphere (Xiao et al., 2015; Wu et al., 2007; Deng et al., 2021), compared to semi-empirical power-law
functions, this parameterization of particle formation rates would be more effective in predicting the NPF occurrence in urban
atmosphere. Additionally, the particle formation rates from other nucleation mechanisms should also be suppressed by high
CS, which needs further exploration and parameterizations. Our methodology of applying pseudo-steady-state assumptions to
kinetic models could be important in reducing computational costs of other SA-amine nucleation systems. For instance,
quantum chemistry calculations also indicate that other basic molecules like trimethylamine and diamines (Jen et al., 2016;
Jen et al., 2014a), might also form relative stable clusters with SA molecules, hence the methodology of parameterizations in
this study could also be extended for them.

**Codes/Data availability**

The codes/data are available upon request from the corresponding author.

**Author Contribution**

Y.L., J.S., B.Z., and J.J. designed the research; J.Z., M.K., and J.J. collected the observational data; Y.L., R.C., and J.J. set up
and tested the parameterization; J.S., B.Z., S.W., and D.G. developed the 3-D model and performed the simulations; Y.L. and
J.S. analyzed the data with the help of R.C., B.Z., and J.J.; M.S. and Y.G. presented important suggestions for the writings;
Y.L., J.S., B.Z., and J.J. wrote the paper with inputs from all co-authors.

**Competing Interests**

Some authors are members of the editorial board of journal *Atmospheric Chemistry and Physics*. The peer-review process was
guided by an independent editor, and the authors have also no other competing interests to declare.

**Acknowledgement**

Financial support from the National Natural Science Foundation of China (22188102, 92044301 and 42275110), Tencent
Foundation through the XPLORER PRIZE and Samsung $PM_{2.5}$ SRP are acknowledged. M. Shrivastava acknowledges the
support from the U.S. Department of Energy (DOE), Office of Science, Office of Biological and Environmental Research
through the Early Career Research Program.



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





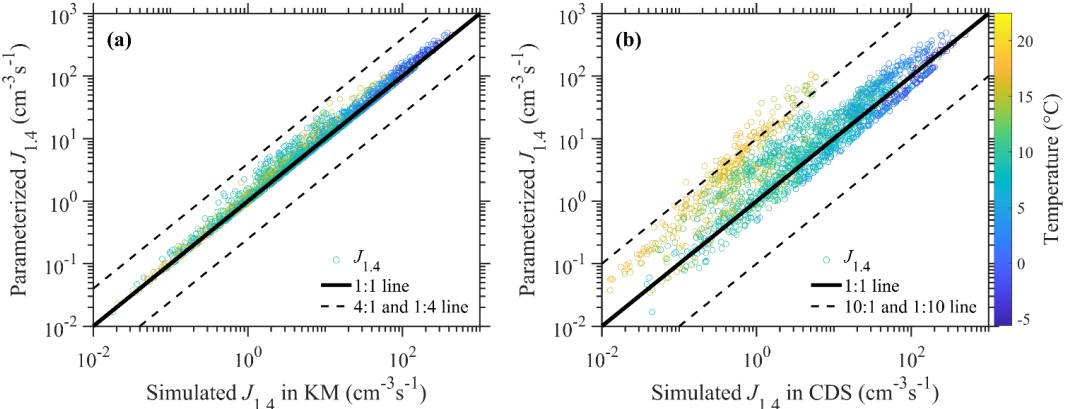

**Figure 1. $J_{1.4}$ Comparison of simplified parameterization method with kinetic model (KM) results (a) and cluster dynamic simulation (CDS) results (b).** The red hollow circles showed the simulation results according to atmospheric observation data. The grey straight line represents the 1:1 line, while the grey dashed line represents the ±50% variation. The circles are colored by the temperatures.



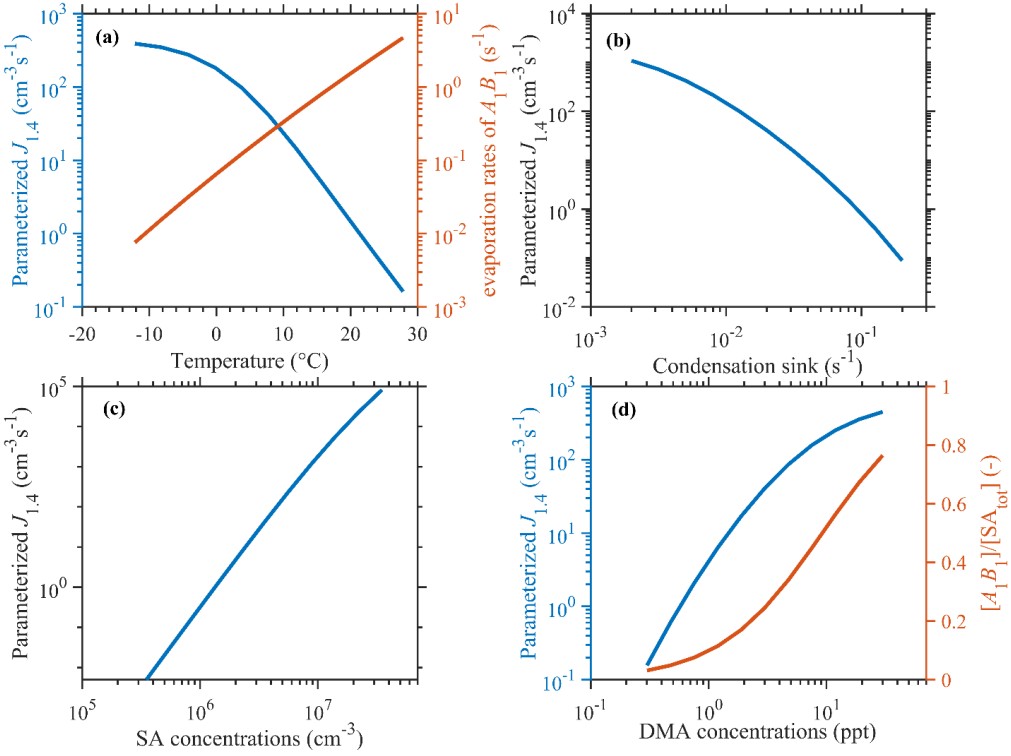


**Figure 2. Dependence of simulation results on varying $T$, $CS$, [DMA], [SA].** The values of fixed parameters are 281K, 0.02
s$^{-1}$, 3 ppt, and 3.5×10$^{6}$ cm$^{-3}$, respectively, as median values during NPF events in our simulation period.





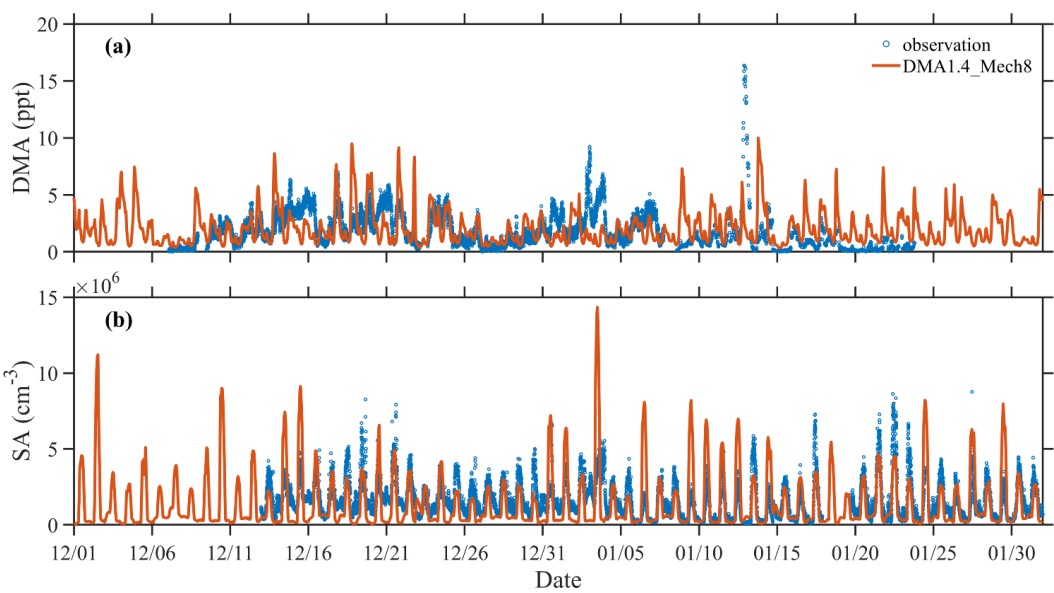


**Figure 3. Comparison of simulated concentrations of DMA (a) and SA (b) with field measurements for wintertime Beijing (December 2018 and January 2019).**



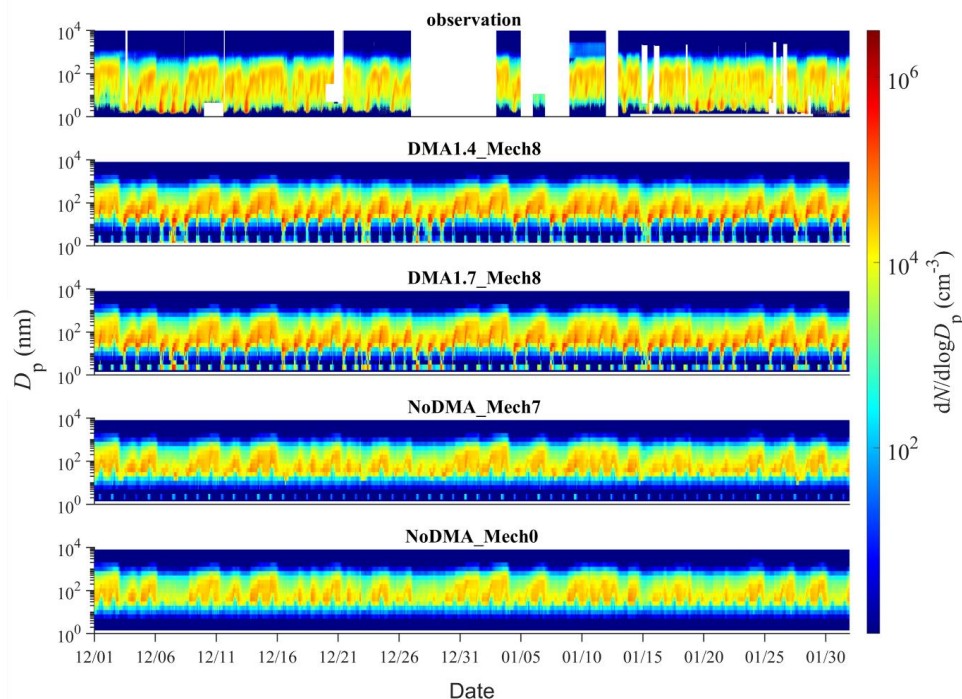


**Figure 4. Comparison of time series of particle number size distribution simulated by various scenarios with the observed one.** Description of four scenarios is detailed in ***Configuration of the Updated WRF-Chem/R2D-VBS Model*** section.








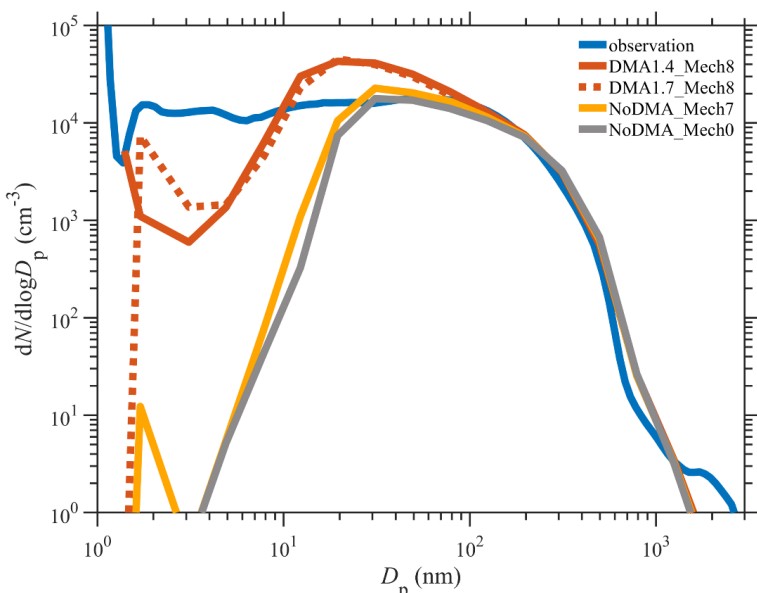


**Figure 5. Comparison of averaged particle number size distribution simulated by various scenarios with the observed one.** Description of four scenarios is detailed in *Configuration of the Updated WRF-Chem/R2D-VBS Model* section.





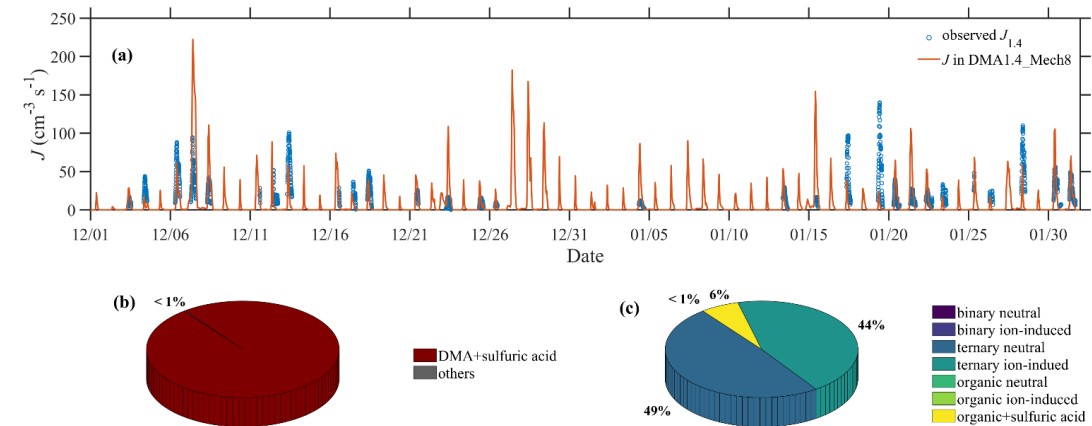


**Figure 6. Comparison of simulated nucleation rates with those derived from field measurements (a), and contribution from different nucleation mechanisms (b) with a special illustration of nucleation pathways other than SA-DMA (c).**