# Peer review of "A dynamic parameterization of sulfuric acid-dimethylamine nucleation and its application in three-dimensional modeling"

_Atmospheric Chemistry and Physics, 2023_

## Author Comment (AC1)

**A parameterization of sulfuric acid-dimethylamine nucleation and its application in three-dimensional modeling**

We are grateful for the referees' comments and these comments have helped us to improve the manuscript. Please find our point-to-point responses below. Comments are shown as blue italic text and the revised texts are shown as "quoted underlined text". In the revised manuscript, the changes are highlighted. The line numbers in the response refer to the revised manuscript without tracked changes.

*Referee 1:*

*The study presents a parameterization of (mostly) previously published detailed modeling of sulfuric-acid- dimethylamine (DMA) new particle formation (NPF) and a modeling study in which the parameterization is applied over Beijing and found to explain aerosol number concentrations there. The parameterization of the detailed models and the method of simulating DMA concentrations are interesting, and useful. The paper could be improved by making the statements in the text more quantitative. I recommend that the paper is within scope of ACP and will be suitable for publication if the following mainly minor comments can be addressed.*

*Major comments*

*1. Need to evaluate condensation sink (CS) timeseries in WRF-chem simulations to show that simulated 1nm aerosol concentrations aren't right for the wrong reasons. Please discuss results with reference to how sensitive the NPF rates are to the CS (Figure 2). While some information might be gained from the banana plots, they are hard to interpret quantitatively, and the size distribution is averaged over the time-period, which might hide significant discrepancies and might also help explain issue in simulating 2-10nm-sized aerosols.*

**Responses:** We thank the reviewer for the valuable comment. With full consideration of Comments 1&8 from Referee 1 and Comment 1 from Referee 2, we acknowledge that the quantitative analysis of the simulation performance of input parameters (CS, [SA] and [DMA]) is lacking. Thus we add the timeseries comparison of CS into Fig. 3 and added correlation coefficient $R^2$ and normalized mean bias (NMB) to evaluate the simulation in the revised manuscript. Also, Fig. S6&S8 are added for further support.

Lines 308-318 in the revised manuscript: "As [DMA], [SA] and CS are key input variables for the $J_{1.4}$ parameterization, we first compare simulated [DMA], [SA] and CS from the DMA1.4_Mech8 scenario with observations (Fig. 3). Generally, there are good consistencies of both mean values and temporal variations, although there are still deviations at certain times. The mean simulated [DMA], [SA], and CS are 1.9 ppt, $1.4 \times 10^6$ cm$^{-3}$, and 0.040 s$^{-1}$, respectively, close to observed values of 2.0 ppt, $1.6 \times 10^6$ cm$^{-3}$, and 0.043 s$^{-1}$. In a quantitative view, the $R^2$ between simulated and observed

[DMA], [SA], and CS are 0.04, 0.37, and 0.40, respectively, while the coefficients during NPF periods increase to 0.12, 0.51, and 0.49. The normalized mean biases (NMBs) between simulated and observed [DMA], [SA], and CS are $4.5 \times 10^{-3}$, -0.22, and -0.36, respectively, while NMBs during NPF periods are -0.40, 0.01, and -0.66. Generally, the simulation of SA concentrations is good, especially during NPF periods with intense nucleation. We note that the correlation between simulated and observed DMA concentration is lower, which may be attributed to the large uncertainty of the diurnal variation of amine emission. Nevertheless, during NPF periods, the differences between the observed DMA concentration (0.78±0.60 ppt) and our simulation (1.10±0.60 ppt) is relatively small."

Lines 352-356 in the revised manuscript: "The observation-simulation comparison of averaged PNSDs is further conducted for individual NPF days. As shown in Fig. S8, the simulated PNSDs on all NPF days follow a similar pattern as the two-month-averaged one in Fig. 6, indicating that nucleation in each simulated NPF day is accompanied by subsequent rapid growth. The difference in the concentration of 2-10 nm particles between observation and simulation is therefore a common feature on various days and is probably attributed to the simplified assumption in particle growth simulation."

*2. Authors need to follow ACP open data standards: https://www.atmospheric-chemistry-and-physics.net/policies/data_policy.html. It is no longer sufficient to say "data are available from the authors on request'. Post simulation output and all code and data needed to reproduce figures on a repository, at a minimum.*

**Responses:** Thanks, we accepted this suggestion. We have posted our supporting data on Zenodo. The link is https://github.com/laoyeyelao/new-SA-DMA-parameterization.git;

Lines 448-449 in the revised manuscript: " The simulation output data and codes needed for figure reproduction have been posted on Github. The link is https://github.com/laoyeyelao/new-SA-DMA-parameterization.git."

*Minor comments*

*3. The introduction refers to much of the relevant literature on NPF parameterizations but does not review prior modeling work in Beijing. Also need to discuss complementary modeling of SA-DMA clustering by Liu et al 2021, https://www.pnas.org/doi/epdf/10.1073/pnas.2108384118*

**Responses:** Thanks, we have referred to a number of modeling work focusing on NPF in Beijing (Chen et al., 2019; Chen et al., 2021). We have also included some studies

showing the role of other molecules in enhancing SA-DMA nucleation (Liu et al., 2021; Glasoe et al., 2015; Wang et al., 2021).

Lines 47-48 in the revised manuscript:"Furthermore, other molecules, such as $HNO_3$ and $NH_3$, could enhance the SA-DMA nucleation under certain conditions."

Lines 49-50 in the revised manuscript:"Although a few previous 3-D simulation studies have simulated NPF events in polluted urban atmospheres such as Beijing, they didn't take the SA-amine nucleation into account."

*4. Is there any dependence of the SA-DMA NPF rates on relative humidity?*

**Responses:** As shown in the following figure (Fig. R1), there is no clear dependence of observed $J_{1.4}$ on relative humidity, and NPF events seldom occur on high RH (e.g., 50% or higher) days in Beijing. A possible reason is that there is usually higher CS on high RH days, which has a suppressing effect on nucleation. Another reason can be that if water could influence SA-DMA nucleation rates, the relationship should be found between $J_{1.4}$ and absolute humidity. But whether water has a contribution to SA-DMA nucleation is still not known yet because it is within the uncertainties of both measurements and quantum chemistry simulation.

[Figure]

Fig. R1 The observed $J_{1.4}$ and the corresponding RH during the observation period

*5. The evaporation rate of 3.33s-1 is different to the one in Kuerten et al (2018), 0.1s-1 and likely other studies. Why?*

**Responses:** Temperature difference is the key reason. We use 298.15 K as a reference temperature and the evaporation rate is highly sensitive to $T$. If we let $T$ be 278 K (Kurten et al., 2018), the rate would be 0.07 s$^{-1}$, consistent with the results derived from experiments in Kurten et al. (2018).

*6. What about synergistic effects involving ammonia or nitric acid (Glasoe et al, 2015; Liu et al 2021) and what about the possible role of amines other than DMA or even malic acid (Liu et al, Phys Chem Chem Phys, 2022 10.1039/D2CP03551K).*

**Responses:** Thanks, we think this question could be divided into two parts. For acids or other non-basic molecules that were reported to enhance the SA-DMA nucleation in certain conditions, we acknowledge that it will be better if the enhancing effect could be considered accurately in the parameterization. However, these studies usually focus on the cluster stability under certain conditions, but such effects have not been proven to play an important role in real atmospheric conditions. Also considering that the observational data only reported SA-amine clusters to be highly correlated with NPF events (Yao et al., 2018; Yin et al., 2021; Cai et al., 2021; Cai et al., 2022), we focused on the SA-DMA nucleation in this study. Future studies can be performed to conduct more experimental or quantum chemistry studies based on real atmospheric temperatures and molecule concentration ranges to explore the co-effect of SA-DMA with other molecules.

For basic molecules like trimethylamine (TMA), if the clusters of sulfuric acid and them are strong enough to initialize NPF events, our methodology could also be extended to set up the parameterizations. For instance, the basic molecules could be treated as equivalent DMA concentrations to simplify the parameterization. However, we do acknowledge that the co-existence of different basic molecules with DMA is also a question requiring further studies.

Lines 443-446 in the revised manuscript:", or the basic molecules could also be treated as equivalent DMA concentrations. Note that although some studies have revealed that SA-DMA nucleation could also be enhanced by adding other molecules in certain conditions, quantitative analysis of these effects in relevant atmospheric conditions is still lacking, thus in this study, we set up this parameterization only based on SA-DMA binary nucleation."

*7. Sections 3.1 and 3.2 are too brief and qualitative: need to quantify the biases in the parameterized J rates versus the KM and CDS models using normalized mean bias and R^2.*

**Responses:** Thanks for this comment, and we revise Fig.1 and the manuscript.

Line 263 in the revised manuscript:", with the correlation coefficient ($R^2$) and normalized mean bias (NMB) of 0.9297 and 0.16, respectively."

Lines 267-268 in the revised manuscript: "The $R^2$ and NMB of the simulated $J_{1.4}$ between this parameterization and CDS are 0.7244 and 0.29, respectively."

*8. Section 3.3: what are the R^2 values between the measured and simulated SA and DMA concentrations? Daytime SA is usually simulated within a factor 2 of measurements, which looks good, and the measurement uncertainty is likely close to a factor of 2 if not higher, so you can't be expected to do much better than this – but a factor of 2 in SA is a factor of 16 in SA-DMA nucleation if the power law is 4. Doesn't this introduce an important uncertainty in the simulated aerosol number concentration?*

**Responses:** The $R^2$ of [SA] and [DMA] have been shown in the manuscript (See responses of comment 1) and the influence of high sensitivity to [SA] and CS is discussed in the revised manuscript.

Lines 318-322 in the revised manuscript: "For [SA] and CS, to which $J_{1.4}$ are most sensitive, we compare the timeseries of simulated and observed $[SA]^4/CS^2$ (based on the approximate dependence of $J_{1.4}$ on [SA] and CS, as shown in Fig. 2)during NPF periods to show the deviations of the combination of these two input parameters (Fig. S6). Generally, in most nucleation events, the simulated values would not deviate from the observed values by over an order of magnitude. This indicates the validity of the comprehensive representation of input parameters in the model."

*9. Section 3.4: In Figure 6, it looks like only about half of the NPF events simulated actually happened. It would be good to put some more precise numbers on this in the text. And in Figure 4, there seem to be too many Aitken mode aerosols most of the time. Doesn't this suggest the SA-DMA NPF mechanism is too strong? What could cause these biases?*

**Responses:** Firstly, the observational data of certain periods are missing and the lower number of observed NPF events in Fig. 6 compared to the simulated ones is largely attributed to the absence of observational data. To avoid confusions, we have hidden the simulated values for these periods. Also, we acknowledge that some periods have very low observed nucleation rates but higher simulated values, which is a result of deviations between the simulated input parameters and the observed input parameters ([SA], [DMA], CS, etc.). Nevertheless, the simulation has successfully captured the magnitude and variation of nucleation rates.

*10. How frequently were relevant diagnostic variables output from WRF-chem?*

**Responses:** The relevant diagnostic variables were output every one hour.

*11. **Figure 1:** the dashed line represents a factor of four variation in a) and an order of magnitude in b). There is no grey dashed line for +/-50% variation.*

**Responses:** Thanks for this comment, and we revise Fig.1 accordingly.

*12. **Line 308/Figure S6:** Dunne et al (2016) included SA-DMA nucleation in their model but did not present it as part of their main analysis. Was only SA-H2O and SA-NH3-H2O nucleation from CLOUD included in this comparison, or were all mechanisms included? Please clarify.*

**Responses:** In Dunne et al.'s study, DMA-SA nucleation can contribute 6%-17% to nucleation under the height of 500 m (Dunne et al., 2016). This low contribution compared to this study can be attributed to two aspects: (1) the lower DMA concentrations simulated in Dunne's study (organic amine emissions from many ammonia emission categories such as crops and fertilizers are ignored); (2) the difference in the parameterization schemes (at the DMA concentrations simulated in this study, the simulated nucleation rate using Dunne et al.'s parameterization is still about an order of magnitude lower). Therefore, results from Dunne et al.'s study may underestimate the intensity of DMA-SA nucleation. As described in Section 2.2, the parameterizations of all seven mechanisms other than DMA-SA nucleation derived from CLOUD chamber are incorporated in our model, including SA-$H_2O$ and SA-$NH_3$-$H_2O$ nucleation.

*13. All the Supplementary figures should be explicitly referred to in the text, or removed – but it may be enough to change S9-S16 to S9-S18 on line 362.*

**Responses:** Thanks, we made the changes accordingly..

*14. Would be good to try to link sensitivity studies more closely to observed biases in the results – discuss how uncertainties in X could lead to biases in Y etc.*

**Responses:** Thanks, according to the dependence of input parameters reported in Section 3.2 and the sensitivity test in Section 3.5, we find that the sensitivities of $J_{1.4}$ to [SA] and CS are higher than that to [DMA]. The $R^2$ and NMB of input parameters showed that simulated [SA], [DMA] and CS are all on average lower than observed values. The simulations of [SA] and CS are closer to observations, compared with those of [DMA].

For [SA] and CS, we note that the simulation biases of them could also influence each other. Thus, we test our simulations of $[SA]^4/CS^2$ (which is approximately proportional to parameterized $J_{1.4}$), and find that the simulations are not so different from the

observations. This implies that even if the simulations could be modified with perfectly accurate simulations of both [SA] and CS, the modification would not change the simulated $J_{1.4}$ so much.

For [DMA], we found the mean observed [DMA] is roughly 1.4 times of the mean simulated values. If we compare the observed $J_{1.4}$ with those simulated in SenDMA2 (simulations with [DMA] doubled, even higher than mean observed [DMA]), we can see that the simulated $J$ are on average higher than observations, but still comparable for some specific cases. This indicates that the lower simulated [DMA] might be a reason for lower simulated $J$, but would not bring much bias.

Lines 318-322 in the revised manuscript: "For [SA] and CS, to which $J_{1.4}$ are most sensitive, we compare the timeseries of simulated and observed $[SA]^4/CS^2$ (based on the approximate dependence of $J_{1.4}$ on [SA] and CS, as shown in Fig. 2)during NPF periods to show the deviations of the combination of these two input parameters (Fig. S6). Generally, in most nucleation events, the simulated values would not deviate from the observed values by over an order of magnitude. This indicates the validity of the comprehensive representation of input parameters in the model."

Revised manuscript, lines 398-402: "In DMA0.5, the simulated $J$ are lower than those observed in almost all cases. In contrast, although the simulated $J$ in DMA2 is on average higher than observations, they are comparable in some specific cases. Considering that during NPF cases, the observed [DMA] are averagely 1.4 times higher than those simulated in DMA1.4_Mech8, we propose that the slight underestimation of DMA concentrations in this case might be the reason for underestimation in $J$ in some cases."

*15. Numerous small grammatical mistakes, e.g. missing "the", use of "largely" to mean "greatly", new words such as "majorly", consistency between singular and plural verb forms (e.g 'the clusters… is…') throughout should be fixed before publication.*

**Responses:** Thanks, we accepted this comment and revised them in the manuscript.

*Referee 2*

*Increasing evidences indicate that sulfuric acid (SA)-driven new particle formation (NPF) enhanced by dimethylamine (DMA) could be the key nucleation mechanism in the polluted urban atmosphere while it has not been well represented in chemical transport models. This study provides a simplified dynamic DMA-SA nucleation parameterization with explicit consideration of the coagulation/condensation sink (CS). Combined with the comprehensive representation of sources and sinks of gaseous precursors, the updated WRF-Chem model well explains NPF events and particle number concentrations in wintertime for Beijing. The manuscript is well organized and the new nucleation modeling methodology has the potential to be*

*widely used in other models. I recommend the publication of the manuscript in ACP if the following minor comments are addressed.*

*1. Both the nature of simplified parameterization and the results of 3-D modeling demonstrate a high dependence of nucleation rate and occurrence of NPF events on CS. Hence, the authors need to evaluate the model performance on CS simulation.*

**Responses:** We thank the reviewer for the valuable comment. With full consideration of Comments 1&8 from Referee 1 and Comment 1 from Referee 2, we acknowledge that the quantitative analysis of the simulation performance of input parameters (CS, [SA] and [DMA]) is lacking. Thus we add the timeseries comparison of CS into Fig. 3 and added correlation coefficient $R^2$ and normalized mean bias (NMB) to evaluate the simulation in the revised manuscript. Also, Fig. S6&S8 are added for further support.

Lines 308-318 in the revised manuscript: "As [DMA], [SA] and CS are key input variables for the $J_{1.4}$ parameterization, we first compare simulated [DMA], [SA] and CS from the DMA1.4_Mech8 scenario with observations (Fig. 3). Generally, there are good consistencies of both mean values and temporal variations, although there are still deviations at certain times. The mean simulated [DMA], [SA], and CS are 1.9 ppt, $1.4 \times 10^6$ cm$^{-3}$, and 0.040 s$^{-1}$, respectively, close to observed values of 2.0 ppt, $1.6 \times 10^6$ cm$^{-3}$, and 0.043 s$^{-1}$. In a quantitative view, the $R^2$ between simulated and observed [DMA], [SA], and CS are 0.04, 0.37, and 0.40, respectively, while the coefficients during NPF periods increase to 0.12, 0.51, and 0.49. The normalized mean biases (NMBs) between simulated and observed [DMA], [SA], and CS are $4.5 \times 10^{-3}$, -0.22, and -0.36, respectively, while NMBs during NPF periods are -0.40, 0.01, and -0.66. Generally, the simulation of SA concentrations is good, especially during NPF periods with intense nucleation. We note that the correlation between simulated and observed DMA concentration is lower, which may be attributed to the large uncertainty of the diurnal variation of amine emission. Nevertheless, during NPF periods, the differences between the observed DMA concentration ($0.78 \pm 0.60$ ppt) and our simulation ($1.10 \pm 0.60$ ppt) is relatively small."

Lines 334-335 in the revised manuscript: "The results were also validated through comparison between the timeseries of the simulated and observed CS (Fig. 3c). "

*2. In section 3.1, "characteristic equilibrium time" and "data collection time interval" are used to test the reasonability of pseudo-steady-state assumptions in SA-DMA parameterization. A more specific description of these terms is suggested for better readability.*

**Responses:** Thanks, we revise the "characteristic equilibrium time" and "data collection time interval" into "e-folding time of cluster formation" and "time interval of observational data (30 min in this study)" for better readability.

Line 251 in the revised manuscript:" e-folding time of cluster dynamics ($\tau$) in the kinetic simulation."

Revised Manuscript Lines 251-252:" time interval of observational data (30 min in this study)."

*3. Various parameterizations of DMA-SA nucleation are reviewed in the introduction, however, previous 3-D modeling works concerning NPF in Beijing are not mentioned.*

**Responses:** Thanks, we have referred to a number of modeling work focusing on NPF in Beijing (Chen et al., 2019; Chen et al., 2021). We have also included some studies showing the role of other molecules in enhancing SA-DMA nucleation (Liu et al., 2021; Glasoe et al., 2015; Wang et al., 2021).

Lines 47-48 in the revised manuscript:"Furthermore, other molecules, such as $HNO_3$ and $NH_3$, could enhance the SA-DMA nucleation under certain conditions."

Lines 49-50 in the revised manuscript:" Although a few previous 3-D simulation studies have simulated NPF events in polluted urban atmospheres such as Beijing, they didn't take the SA-amine nucleation into account "

*4. Some molecular modeling works show that the growth pathways from precursor molecules to ~1.4 nm particles may be altered with changing precursor concentration and atmospheric conditions (T and CS). The underlying uncertainty on growth pathways should be discussed since a fixed one is used under all conditions.*

**Responses:** Thanks, we have revised the **Method** part to clarify this assumption.

[revised manuscript text omitted]

---

## Editor Decision (ED1)

Suggested Revision to the Title:

Add "Dynamic" since loss processes are an important element of this new parameterization.
"A Dynamic Parameterization of Sulfuric Acid-Dimethylamine…"
Or
"A Dynamics-Based Parameterization of Sulfuric Acid-Dimethylamine…"

Suggested Revisions to the Abstract:

Reword first sentence to better link NPF and source of particles, and replace "diverse" to more strongly state that SA has been linked to NPF in all studied environments:

Sulfuric acid (SA) is a governing gaseous precursor for atmospheric new particle formation (NPF), a major source of ultrafine particles, in environments studied around the world.

Replace "atmosphere" with "atmospheres" to indicate more than one urban atmosphere; change throughout manuscript when describing urban atmospheres generally:

"polluted urban atmospheres with a high condensational sink"

Replace "they" with "these loss processes" to more concisely refer to the loss and not the general representation of the contribution of these clusters to NPF:

Coagulation scavenging and cluster evaporation are dominant sink processes of SA-amine clusters in urban atmospheres, yet these loss processes are not quantitatively included in the present parameterizations of SA-amine nucleation.

Reconsider "would be able to reproduce" (line 29) and "would improve the performance" (line 36). When the statement has been demonstrated, it is recommended to state this more conclusively. When the statement has not been demonstrated, it is recommended to state this more hypothetically. For example:

Compared with previous SA-DMA nucleation parameterizations, this new parameterization was able to reproduce the dependences of particle formation rates on temperature and CS.

Representing these processes is thus likely to improve model performance in particle source apportionment and quantification of aerosol effects on air quality, human health, and climate.

It is recommended to revisit the use of "would" throughout the manuscript. It suggests speculation rather than observation or deduction. For example (from section 2.1):

Based on previous studies, under atmospheric conditions, variations of precursor concentrations, temperature and CS do not result in large deviations to the main pathway. Simulations under different [SA], [DMA], and temperatures have shown that the main pathway was similar under the different conditions studied (Olenius et al.,

[revised manuscript text omitted]

I have modified and included this sentence at the end, but I am not sure what it means:
*The improved simulations of particle size distributions also provide more evidence for quantitatively evaluate the environmental and health effect of ultrafine particles.*

The fact that NPF is important in urban environments does support evaluating the environmental and health effects of ultrafine particles, but I'm not sure how this relates to a new model parameterization. The measurements themselves demonstrate that such particles/events exist. Maybe you are suggesting that if we have more accurate model simulations, we can more readily use particle size distributions as a metric since we won't have to rely on measurements.

Suggestion to move this to the methods section:
Although some studies have revealed that SA-DMA nucleation could also be enhanced by adding other molecules in certain conditions, quantitative analysis of these effects in

relevant atmospheric conditions is still lacking, thus in this study, we set up this parameterization only based on SA-DMA binary nucleation.

---

## Author Response (AR2)

**Responses to Editor's Comments on Manuscript acp-2023-15**
**A parameterization of sulfuric acid-dimethylamine nucleation and its application
in three-dimensional modeling**

We are grateful for the editor's comments and these comments have helped us to improve the manuscript. Please find our point-to-point responses below. Comments are shown as blue italic text and the revised texts are shown as "quoted underlined text". In the revised manuscript, the changes are highlighted. The line numbers in the response refer to the revised manuscript without tracked changes.

*Referee 1:*

*Suggested Revision to the Title:*
*Add "Dynamic" since loss processes are an important element of this new parameterization.*
   *"A Dynamic Parameterization of Sulfuric Acid-Dimethylamine…"*
*Or*
   *"A Dynamics-Based Parameterization of Sulfuric Acid-Dimethylamine…"*
Response: Thanks for the comment. We have changed the title to "A Dynamic Parameterization …".

*Suggested Revisions to the Abstract:*
*Reword first sentence to better link NPF and source of particles, and replace "diverse" to more strongly state that SA has been linked to NPF in all studied environments: Sulfuric acid (SA) is a governing gaseous precursor for atmospheric new particle formation (NPF), a major source of ultrafine particles, in environments studied around the world.*
*Replace "atmosphere" with "atmospheres" to indicate more than one urban atmosphere; change throughout manuscript when describing urban atmospheres generally: "polluted urban atmospheres with a high condensational sink" Replace "they" with "these loss processes" to more concisely refer to the loss and not the general representation of the contribution of these clusters to NPF: Coagulation scavenging and cluster evaporation are dominant sink processes of SAamine clusters in urban atmospheres, yet these loss processes are not quantitatively included in the present parameterizations of SA-amine nucleation.*
*Reconsider "would be able to reproduce" (line 29) and "would improve the performance" (line 36). When the statement has been demonstrated, it is recommended to state this more conclusively. When the statement has not been demonstrated, it is recommended to state this more hypothetically. For example: Compared with previous SA-DMA nucleation parameterizations, this new parameterization was able to reproduce the dependences of particle formation rates on temperature and CS. Representing these processes is thus likely to improve model performance in particle source apportionment and quantification of aerosol effects on air quality, human health, and climate.*
*It is recommended to revisit the use of "would" throughout the manuscript. It suggests speculation rather than observation or deduction. For example (from section 2.1):*

[revised manuscript text omitted]

Response: We have revised the expressions as the editor suggested.
Lines 119-121 in the revised manuscript:” Although some studies have revealed that SA-DMA nucleation could also be enhanced by adding other molecules in certain conditions, quantitative analysis of these effects in relevant atmospheric conditions is still lacking, thus in this study, we set up this parameterization only based on SA-DMA binary nucleation.”